# Black-Box Optimization with Local Generative Surrogates

**Sergey Shirobokov**[*]
Department of Physics
Imperial College London
United Kingdom
s.shirobokov17@imperial.ac.uk

**Vladislav Belavin**[*]
National Research University
Higher School of Economics
Moscow, Russia
vbelavin@hse.ru

**Michael Kagan**
SLAC National Accelerator Laboratory
Menlo Park, CA
United States

**Andrey Ustyuzhanin**
National Research University
Higher School of Economics
Moscow, Russia

**Atılım Güneş Baydin**
Department of Computer Science
Department of Engineering Science
University of Oxford
United Kingdom

## Abstract

We propose a novel method for gradient-based optimization of black-box simulators using differentiable local surrogate models. In fields such as physics and engineering, many processes are modeled with non-differentiable simulators with intractable likelihoods. Optimization of these forward models is particularly challenging, especially when the simulator is stochastic. To address such cases, we introduce the use of deep generative models to iteratively approximate the simulator in local neighborhoods of the parameter space. We demonstrate that these local surrogates can be used to approximate the gradient of the simulator, and thus enable gradient-based optimization of simulator parameters. In cases where the dependence of the simulator on the parameter space is constrained to a low dimensional submanifold, we observe that our method attains minima faster than baseline methods, including Bayesian optimization, numerical optimization, and approaches using score function gradient estimators.

## 1 Introduction

Computer simulation is a powerful method that allows for the modeling of complex real-world systems and the estimation of a system's parameters given conditions and constraints. Simulators drive research in many fields of engineering and science [13] and are also used for the generation of synthetic labeled data for various tasks in machine learning [52, 49, 50, 7]. A common challenge is to find optimal parameters of a simulated system in terms of a given objective function, e.g., to optimize a real-world system's design or efficiency using the simulator as a proxy, or to calibrate a simulator to generate data that match a real-data distribution. A typical simulator optimization problem can be defined as finding $\psi^* = \arg\min_{\psi} \sum_{\boldsymbol{x}} \mathcal{R}(F(\boldsymbol{x}, \psi))$, where $\mathcal{R}$ is an objective we

---

[*]Equal contribution

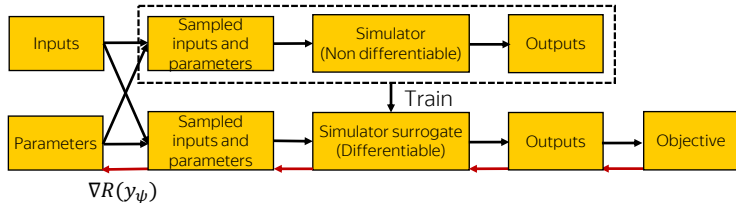

Figure 1: Simulation and surrogate training. *Black:* forward propagation. *Red:* error backpropagation.

would like to minimize and $F$ is a simulator that we take as a black box with parameters $\boldsymbol{\psi} \in \mathbb{R}^n$ and inputs $\boldsymbol{x} \in \mathbb{R}^d$.

In this work, we focus on cases where the simulator and its inputs are stochastic, so that $\boldsymbol{y} = F(\boldsymbol{x}, \boldsymbol{\psi})$ is a random variable $\boldsymbol{y} \sim p(\boldsymbol{y}|\boldsymbol{x}; \boldsymbol{\psi})$, the inputs are $\boldsymbol{x} \sim q(\boldsymbol{x})$, and the objective is expressed as the expectation $\mathbb{E}_{p(\boldsymbol{y}|\boldsymbol{x};\boldsymbol{\psi})}[\mathcal{R}(\boldsymbol{y})]$. Note that the choice of modeling the simulator inputs $\boldsymbol{x}$ as random reflects the situation common in scientific simulation settings, and our methods are equally applicable for the case without stochastic inputs such that $\boldsymbol{y} \sim p(\boldsymbol{y}; \boldsymbol{\psi})$.

In many settings the cost of running the simulator is high, and thus we aim to minimize the number of simulator calls needed for optimization. Such stochastic simulator optimization occurs in an array of scientific and engineering domains, especially in cases of simulation-based optimization relying on Monte Carlo techniques. Examples include optimization for particle scattering simulators [56], radiation transport simulators [4], and molecular dynamics simulation [1]. In some domains, this is also referred to as design optimization [44].

Several methods exist for such optimization, depending on the availability of gradients for the objective function [36]. While gradient-based optimization has been demonstrated to work well for differentiable objective functions [30, 11, 14, 15], non-differentiable objective functions occur frequently in practice, e.g., where one aims to optimize the parameters of a system characterized by a Monte Carlo simulator that may only produce data samples from an intractable likelihood [13]. In such cases, genetic algorithms [44, 5], Bayesian optimization [58, 55], or numerical differentiation [62] are frequently employed. More recently stochastic gradient estimators [41] such as the REINFORCE [65] estimator, have been employed to estimate gradients of non-differentiable functions [60, 12] and subsequently perform gradient-based optimization.

In order to utilize the strengths of gradient-based optimization while avoiding the high variance often observed with score function gradient estimators, our approach employs deep generative models as differentiable surrogate models to approximate non-differentiable simulators, as described in Figure 1. Using these surrogates, we show that we can both approximate the stochastic behavior of the simulator and enable direct gradient-based optimization of an objective by parameterizing the surrogate model with the relevant parameters of the simulator. In high-dimensional parameter spaces, training such surrogates over the complete parameter space becomes computationally expensive. Our technique, which we name *local generative surrogate optimization* (L-GSO), addresses this by using successive local neighborhoods of the parameter space to train surrogates at each step of parameter optimization. Our method works especially well when the parameters, which are seemingly high dimensional, live on a lower dimensional submanifold, as seen in practice in a variety of settings [29].

L-GSO relies primarily on two assumptions: (a) that the objective function is continuous and differentiable, and (b) that the parameters $\psi$ are continuous variables. The first assumption may be relaxed by incorporating the objective into the surrogate.

In Section 2 we describe the L-GSO algorithm. We cover related work in Section 3. In Section 4 we evaluate L-GSO on a set of toy problems and compare it with frequently used methods including numerical differentiation, Bayesian optimization, and score function-based approaches, and present results of a realistic use case in the high-energy physics domain. Section 5 presents our conclusions.

## 2 Method

**Problem Statement** We target an optimization formulation applicable in domains where a simulator characterized by parameters $\boldsymbol{\psi}$ takes stochastic inputs $\boldsymbol{x} \sim q(\boldsymbol{x})$ and produces outputs (observations)

$\boldsymbol{y} \sim p(\boldsymbol{y}|\boldsymbol{x}; \boldsymbol{\psi})$. For example in the case of designing the shape of an experimental device, $\boldsymbol{x}$ may represent random inputs to an experimental apparatus, $\boldsymbol{\psi}$ defines the shape of the apparatus, and $p(\boldsymbol{y}|\boldsymbol{x}; \boldsymbol{\psi})$ encodes the impact of the apparatus on the input to produce observations $\boldsymbol{y}$. A task-specific objective function $\mathcal{R}(\boldsymbol{y})$ encodes the quality of observations and may be optimized over the parameters $\boldsymbol{\psi}$ of the observed distribution. In cases when a simulator $F$ can only draw samples from the distributions $p(\boldsymbol{y}|\boldsymbol{x}; \boldsymbol{\psi})$ the optimization problem can be approximated as

$$\boldsymbol{\psi}^* = \arg\min_{\boldsymbol{\psi}} \mathbb{E}[\mathcal{R}(\boldsymbol{y})] = \arg\min_{\boldsymbol{\psi}} \int \mathcal{R}(\boldsymbol{y}) p(\boldsymbol{y}|\boldsymbol{x}; \boldsymbol{\psi}) q(\boldsymbol{x}) d\boldsymbol{x} d\boldsymbol{y}$$

$$\approx \arg\min_{\boldsymbol{\psi}} \frac{1}{N} \sum_{i=1}^{N} \mathcal{R}(F(\boldsymbol{x}_i; \boldsymbol{\psi}))$$

(1)

where $\boldsymbol{y}_i = F(\boldsymbol{x}_i; \boldsymbol{\psi}) \sim p(\boldsymbol{y}|\boldsymbol{x}; \boldsymbol{\psi})$, $x_i \sim q(\boldsymbol{x})$ and a Monte Carlo approximation to the expected value of the objective function is computed using samples drawn from the simulator. Note that $F$ represents a stochastic process, which may itself depend on latent random variables.

## 2.1 Deep generative models as differentiable surrogates

Given a non-differentiable simulator $F$, direct gradient-based optimization of Eq. 1 is not possible. We propose to approximate $F$ with a learned differentiable model, denoted a surrogate, $\bar{\boldsymbol{y}} = S_\theta(\boldsymbol{z}, \boldsymbol{x}; \boldsymbol{\psi})$ that approximates $F(\boldsymbol{x}; \boldsymbol{\psi})$, where $\boldsymbol{z} \sim p(\boldsymbol{z})$ are latent variables accounting for the stochastic variation of the distribution $p(\boldsymbol{y}|\boldsymbol{x}; \boldsymbol{\psi})$, $\theta$ are surrogate model parameters, and $\bar{\boldsymbol{y}}$ are surrogate outputs. When the samples $\bar{\boldsymbol{y}}$ are differentiable with respect to $\boldsymbol{\psi}$, direct optimization of Eq. 1 can be done with the surrogate gradient estimator:

$$\nabla_{\boldsymbol{\psi}} \mathbb{E}[\mathcal{R}(\boldsymbol{y})] \approx \frac{1}{N} \sum_{i=1}^{N} \nabla_{\boldsymbol{\psi}} \mathcal{R}(S_\theta(\boldsymbol{z}_i, \boldsymbol{x}_i; \boldsymbol{\psi})) \ . $$

(2)

To obtain a differentiable surrogate capable of modeling a stochastic process, $S_\theta$ is defined as a deep generative model whose parameters $\theta$ are learned. Generative model learning can be done independently of the simulator optimization in Eq. 1 as it only requires samples from the simulator to learn the stochastic process. Once learned, the generative surrogate can produce differentiable samples that are used to approximate the integration for the expected value of the objective function. Several types of generative models can be used, including generative adversarial networks (GANs) [21], variational autoencoders [35, 48], or flow-based models [47]. We present results using conditional variants of two recently proposed models, Cramer GAN [8] and the FFJORD continuous flow model [23], to show the independence of L-GSO from the choice of generative model.

## 2.2 Local generative surrogates

The L-GSO optimization algorithm is summarized in Algorithm 1. Using a sample of values for $\boldsymbol{\psi}$ and input samples of $\boldsymbol{x}$, a set of training samples for the surrogate are created from the simulator $F$. The surrogate training step 8 refers to the standard training procedures for the chosen generative model (details on model architectures and hyperparameters are given in Appendix A). The learned surrogate is used to estimate the gradient of the objective function with backpropagation through the computed expectation of Eq. 2 with respect to $\boldsymbol{\psi}$. Subsequently $\boldsymbol{\psi}$ is updated with a gradient descent procedure, denoted *SGD* (stochastic gradient descent) in the algorithm. Due to the use of SGD, an inherently noisy optimization algorithm, the surrogate does not need to be trained until convergence but only sufficiently well to provide gradients correlated with the true gradient that produce useful SGD updates. The level of correlation will control the speed of convergence of the method.

For high-dimensional $\boldsymbol{\psi}$, a large number of parameter values $\boldsymbol{\psi}$ must be sampled to accurately train a single surrogate model. Otherwise the surrogate would not provide sufficiently well estimated gradients over the full parameter space that may be explored by the optimization. Thus optimization using a single upfront training of the surrogate model over all $\boldsymbol{\psi}$ becomes unfeasible. As such, we utilize a trust-region-like approach [59] to train a surrogate model locally in the proximity of the current parameter value $\boldsymbol{\psi}$. We sample new $\boldsymbol{\psi}'$ around the current point $\boldsymbol{\psi}$ from the set $U_\epsilon^{\boldsymbol{\psi}} = \{\boldsymbol{\psi}' : |\boldsymbol{\psi}'_i - \boldsymbol{\psi}_i| \leq \epsilon, \forall i \in \{1, \dots, n\}\}$. Using this local model, a gradient at the current point $\boldsymbol{\psi}$ can be obtained and a step of SGD performed. After each SGD update of $\boldsymbol{\psi}$, a new local surrogate

is trained. As a result, we do not expect domain shift to impact L-GSO as it is retrained at each new parameter point.

In local optimization there are several hyperparameters that require tuning either prior to or dynamically during optimization. One must choose the sampling algorithm for $\boldsymbol{\psi}$ values in the region $U_\epsilon^\psi$ in step 3 of Algorithm 1. In high-dimensional space, uniform sampling is inefficient, thus we have adopted the Latin Hypercubes algorithm [31].[2] One must also choose a proximity hyperparameter $\epsilon$, that controls the size of the region of $\boldsymbol{\psi}_i$ in which a set of $\boldsymbol{\psi}$ values is chosen to train a local surrogate.[3] This hyperparameter is similar to the step size used in numerical differentiation, affecting the speed of convergence as well as the overall behavior of the optimization algorithm; if $\epsilon$ is too large or too small the algorithm may diverge. In this paper we report experimental results with this hyperparameter tuned based on a grid search. The value of $\epsilon = 0.2$ was found to be robust and work well in all experiments except the "three hump problem" which required a slightly larger value of $\epsilon = 0.5$. More details can be found in Appendix C.

---

**Algorithm 1** Local Generative Surrogate Optimization (L-GSO) procedure

---

**Require:** number N of $\boldsymbol{\psi}$, number M of $\boldsymbol{x}$ for surrogate training, number K of $\boldsymbol{x}$ for $\boldsymbol{\psi}$ optimization step, trust region $U_\epsilon$, size of the neighborhood $\epsilon$, Euclidean distance $d$
1: Choose initial parameter $\boldsymbol{\psi}$
2: **while** $\boldsymbol{\psi}$ has not converged **do**
3:     Sample $\boldsymbol{\psi}_i'$ in the region $U_\epsilon^\psi$, $i = 1, \ldots, N$
4:     For each $\boldsymbol{\psi}_i'$, sample inputs $\{\boldsymbol{x}_j^i\}_{j=1}^M \sim q(\boldsymbol{x})$
5:     Sample $M \times N$ training examples from simulator $\boldsymbol{y}_{ij} = F(\boldsymbol{x}_j^i; \boldsymbol{\psi}_i')$
6:     Store $\boldsymbol{y}_{ij}, \boldsymbol{x}_j^i, \boldsymbol{\psi}_i'$ in history $H$
    $i = 1, \ldots, N; j = 1, \ldots, M$
7:     Extract all $\boldsymbol{y}_l, \boldsymbol{x}_l, \boldsymbol{\psi}_l'$ from history $H$, iff $d(\boldsymbol{\psi}, \boldsymbol{\psi}_l') < \epsilon$
8:     Train generative surrogate model $S_\theta(\boldsymbol{z}_l, \boldsymbol{x}_l; \boldsymbol{\psi}_l')$, where $\boldsymbol{z}_l \sim \mathcal{N}(0,1)$
9:     Fix weights of the surrogate model $\theta$
10:    Sample $\bar{\boldsymbol{y}}_k = S_\theta(\boldsymbol{z}_k, \boldsymbol{x}_k; \boldsymbol{\psi})$, $\boldsymbol{z}_k \sim \mathcal{N}(0,1)$, $\boldsymbol{x}_k \sim q(\boldsymbol{x})$, $k = 1, \ldots, K$
11:    $\nabla_\psi \mathbb{E}[\mathcal{R}(\bar{\boldsymbol{y}})] \leftarrow \frac{1}{K} \sum_{k=1}^K \frac{\partial \mathcal{R}}{\partial \bar{\boldsymbol{y}}_k} \frac{\partial S_\theta(\boldsymbol{z}_k, \boldsymbol{x}_k; \boldsymbol{\psi})}{\partial \boldsymbol{\psi}}$
12:    $\boldsymbol{\psi} \leftarrow \text{SGD}(\boldsymbol{\psi}, \nabla_\psi \mathbb{E}[\mathcal{R}(\bar{\boldsymbol{y}})])$
13: **end while**

---

The number of $\boldsymbol{\psi}$ values sampled in the neighborhood is another key hyperparameter. We expect the optimal value to be highly correlated with the dimensionality and complexity of the problem. In our experiments, we examine the quality of gradient estimates as a function of the number of points used for training the local surrogate. We observe that it is sufficient to sample $O(D)$ samples in the neighborhood of $\boldsymbol{\psi}$, where $D$ is the full parameter space dimensionality of $\boldsymbol{\psi}$. In this case, our approach is observed to be similar to numerical optimization which expects $O(D)$ samples for performing a gradient step [62]. However, in cases where the components of $\boldsymbol{\psi}$ relevant for the optimization lie on a manifold of dimensionality lower than $D$, i.e., intrinsic dimensionality $d$ is smaller than $D$, we observe that L-GSO requires $O(d)$ samples for producing a reasonable gradient step, thus leading to the faster convergence of L-GSO than other methods.

Previously sampled data points can also be stored in history and later reused in our local optimization procedure (Algorithm 1). This provides additional training points for the surrogate as the optimization progresses and results in a better surrogate model and, consequently, better gradient estimation. The ability of L-GSO to reuse previous samples is a crucial point to reduce the overall number of calls to the simulator. This procedure was observed to aid both FFJORD and GAN models to attain the minimum faster and to prevent the optimization from diverging once the minimum has been attained.

A benefit of our approach in comparison with numerical gradient estimation is that a deep generative surrogate can learn more complex approximations of the objective function than a linear approximation, which can be beneficial to obtain gradients for surfaces with high curvature. We believe that this is mainly due to the implicit regularization afforded by generative neural network architectures [67]. In addition, using generative neural networks as surrogates provides other potential benefits such as Hessian estimation, that may be used for second-order optimization algorithms and/or uncertainty estimation, and possible automatic determination of a low-dimensional parameter manifold. While we do not have theoretical guarantees for the convergence of L-GSO, empirically we do not observe bias in the estimated gradients when performing gradient descent as seen in Section 4.

# 3 Related work

Our work intersects with both simulator optimization and likelihood-free inference. In terms of simulator optimization, our approach can be compared to Bayesian optimization (BO) with Gaussian process based surrogates [58, 55] and numerical derivatives [62]. In comparison with BO, our optimization approach makes use of gradients of the objective surface approximated using a surrogate, which can result in faster and more robust convergence in multidimensional spaces with high curvature (see the Rosenbrock example in Figure 2c). Importantly, our approach does not require covariance matrix inversion which costs $O(n^3)$ where $n$ is a number of observations, thus making it impractical to compute in high dimensional spaces. To make BO scalable, authors often make structural assumptions on the function that may not hold generally. For example, references [64, 16, 19, 66] assume a low-dimensional linear subspace that can be decomposed in subsets of dimensions. In addition, BO may require the construction of new kernels [22], such as [43] which proposes a cylindrical kernel that penalizes close to boundary points in the search space. Our approach does not make structural assumptions of the parameter manifold or assumptions on the locality of the optimum. While our approach does not require the design of a task-specific kernel, it does require the selection of a surrogate model structure. The method of reference [17] maintains multiple local models simultaneously and samples new data points via an implicit multi-armed bandit approach.

In likelihood-free inference, one aim is to estimate the parameters of a generative process with neither a defined nor tractable likelihood. A major theme of this work is posterior or likelihood density estimation [45, 39, 25]. Our work is similar in its use of sequentially trained generative models for density estimation, but we focus on optimizing any user-defined function, and not specifically a likelihood or posterior, and our sequential training is used to enable gradient estimation rather than updating a posterior. Other recent examples of work that address this problem include [38, 24, 52]. While the first reference discusses non-differentiable simulators, it targets tuning simulator parameters to match a data sample and not the optimization of a user-defined objective function.

Non-differentiable function optimization using score function gradient estimators is explored in [24, 52]. These approaches are applicable to our setting and we provide comparisons in our experiments. Instead of employing the simulator within the computation of score function gradient estimate, our approach builds a surrogate simulator approximation to estimate gradients.

Using deep generative models for optimization problems has been explored in [26, 10]. These approaches focus on posterior inference of the parameters given observations, whereas we focus on direct gradient based objective optimization. These algorithms also assume that it is not costly to sample from the simulator to perform posterior estimation, while sampling from the simulator is considered to be the most computationally expensive step to be minimized in our approach. In addition, [20] use generative models to learn a latent space where optimization is performed, rather than direct optimization over the parameter space of a surrogate forward model as in our approach. Reference [18] aims to find optimal parameters of a generator network while approximating a fixed black box metric, whereas our method optimizes the parameters of a black box function itself.

The success of L-GSO in high-dimensional spaces with low-dimensional submanifolds is consistent with findings on the approximation quality of deep networks and how they adapt to the intrinsic dimension of data [61, 42, 53]. Although restricted to specific function classes, these results provide bounds on approximation quality that reduce with small intrinsic dimension. They are suggestive of the benefits of deep generative models over submanifolds that we empirically observe.

# 4 Experiments

We evaluate L-GSO on five toy experiments in terms of the attained optima and the speed of convergence, and present results in a physics experiment optimization. As simulation is assumed to be the most time consuming operation during optimization, the speed of convergence is measured by the number of simulator calls. The toy experiments, defined below, were chosen to explore low- and high-dimensional optimization problems, and those with parameters on submanifolds. Non-stochastic versions of the experiments are established benchmark functions in the optimization literature [32].

**Probabilistic Three Hump Problem**    We aim to find the 2-dimensional $\boldsymbol{\psi}$ that optimizes:

$$\boldsymbol{\psi}^* = \arg\min_{\boldsymbol{\psi}} \mathbb{E}[\mathcal{R}(y)] = \mathbb{E}[\sigma(y - 10) - \sigma(y)], \text{ s.t.}$$

$$y \sim \mathcal{N}(y; \mu_i, 1), i \in \{1, 2\}, \quad \mu_i \sim \mathcal{N}(x_i h(\boldsymbol{\psi}), 1), \quad x_1 \sim \mathrm{U}[-2, 0], \quad x_2 \sim \mathrm{U}[2, 5] \quad (3)$$

$$\mathrm{P}(i = 1) = \frac{\psi_1}{||\boldsymbol{\psi}||_2} = 1 - \mathrm{P}(i = 2), \quad h(\boldsymbol{\psi}) = 2\psi_1^2 - 1.05\psi_1^4 + \psi_1^6/6 + \psi_1\psi_2 + \psi_2^2$$

**Rosenbrock Problem**    In the N-dimensional Rosenbrock problem we aim to find $\boldsymbol{\psi}$ that optimizes:

$$\boldsymbol{\psi}^* = \arg\min_{\boldsymbol{\psi}} \mathbb{E}[\mathcal{R}(y)] = \arg\min_{\boldsymbol{\psi}} \mathbb{E}[y], \text{ s.t.}$$

$$y \sim \mathcal{N}\left(y; \sum_{i=1}^{n-1} \left[(\psi_i - \psi_{i+1})^2 + (1 - \psi_i)^2\right] + x, 1\right), \quad x \sim \mathcal{N}(x; \mu, 1), \quad \mu \sim \mathrm{U}[-10, 10] \quad (4)$$

**Submanifold Rosenbrock Problem**    To address problems where the parameters lie on a low-dimension submanifold, we define the submanifold Rosenbrock problem, with a mixing matrix $A$ to project the parameters onto a submanifold.[4] In our experiments $\boldsymbol{\psi} \in \mathbb{R}^{100}$ and $A \in \mathbb{R}^{10 \times 100}$ has full row rank. Prior knowledge of $A$ or the submanifold dimension is not used in the surrogate training. The optimization problem is thus defined as:

$$\boldsymbol{\psi}^* = \arg\min_{\boldsymbol{\psi}} \mathbb{E}[\mathcal{R}(y)] = \arg\min_{\boldsymbol{\psi}} \mathbb{E}[y], \text{ s.t. } y \sim \mathcal{N}\left(y; \sum_{i=1}^{n-1} \left[(\psi_i' - \psi_{i+1}')^2 + (1 - \psi_i')^2\right] + x, 1\right)$$

$$\boldsymbol{\psi}' = A \cdot (\boldsymbol{\psi}[\text{mask}]), \quad x \sim \mathcal{N}(x; \mu, 1), \quad \mu \sim \mathrm{U}[-10, 10] \quad (5)$$

**Nonlinear Submanifold Hump Problem**    This experiment explores non-linear submanifold embeddings of the parameters $\psi$. The embedding is through $\hat{\psi} = B\tanh(A\psi)$, where $\psi \in \mathbb{R}^{40}$, $A \in \mathbb{R}^{16 \times 40}$, $B \in \mathbb{R}^{2 \times 16}$, with $A$ and $B$ generated as in the submanifold Rosenbrock problem.[4] The intrinsic dimension of the parameter space is equal to two. The embedded parameters $\hat{\psi}$ are used in place of $\psi$ in the three hump problem definition. We use this example to also explore the number of parameter points needed per surrogate training for effective optimization on a submanifold.

**Neural Network Weights Optimization Problem**    In this problem, we optimize neural network weights for regressing the Boston house prices dataset [28]. As discussed by [37], neural networks are often overparameterized, thus having a smaller intrinsic dimension than the full parameter space dimension. In this experiment we explore the optimization capability of L-GSO over the number of parameter space points needed per surrogate training, and, indirectly, the intrinsic dimensionality of the problem. The problem is defined as:

$$\boldsymbol{\psi}^* = \arg\min_{\boldsymbol{\psi}} \mathbb{E}[\mathcal{R}(y)] = \arg\min_{\boldsymbol{\psi}} \sqrt{\frac{1}{N}\sum_{i=1}^{N}(y - y_{\text{true}})^2}, \text{s.t. } y = \mathrm{NN}(\boldsymbol{\psi}, \boldsymbol{x}), \boldsymbol{x} \sim \{\boldsymbol{x}_i\}_{i=1}^{506} \quad (6)$$

**Baselines**    We compare L-GSO to: Bayesian optimization using Gaussian processes with cylindrical kernels [43], which we denote "*BOCK*", and with radial basis function kernels [55], which we denote "*BO-RBF*", numerical differentiation with gradient descent (referred to as numerical optimization), and guided evolutionary strategies [40] (referred to as "*CMA-ES*").

We also compare with score function-based optimization approaches. In "Learning to Simulate" [52], in order to apply score function gradient estimation of a non-differentiable simulator, the authors introduce a policy over the simulator parameters, $p(\psi|\eta)$, and optimize the policy parameters $\eta$. We denote this method "LTS". Reference [24] introduces the LAX gradient which uses an optimized control variate to reduce the variance of the score function gradient estimator. We cannot apply this method directly, as the authors optimize objectives of form $L(\psi) = \mathbb{E}_{p(y|\psi)}[f(y)]$ and use knowledge of the gradient $\nabla_\psi \log p(y|\psi)$, and in our setting with a non-differentiable simulator this

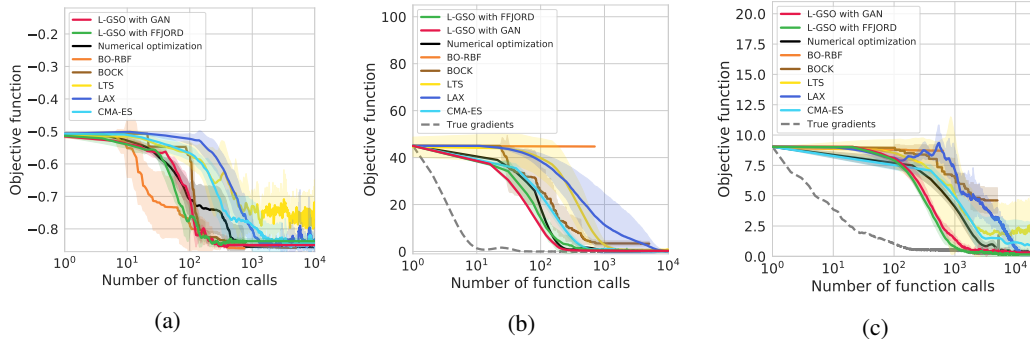

(a)             (b)             (c)

Figure 2: The objective function value on the toy problems for baselines and our method. (a) Three hump problem, (b) Rosenbrock problem [51] in 10 dimensions, initial point is $\vec{2} \in \mathbb{R}^{10}$. (c) Submanifold Rosenbrock Problem in 100 dimensions, initial point is $\vec{2} \in \mathbb{R}^{100}$. True gradients are shown in gray dashed curves when available. Shaded region corresponds to $1\sigma$ confidence intervals.

gradient is not available. Following [52], we treat the objective as a function of the parameters, $f(\psi) = \mathbb{E}_{p(\boldsymbol{y}|\boldsymbol{x};\boldsymbol{\psi})}[\mathcal{R}(\boldsymbol{y})]$, introduce a Gaussian policy $p(\psi|\mu, \sigma) = N(\psi|\mu, \sigma)$, and optimize the objective $L(\mu, \sigma) = \mathbb{E}_{p(\psi|\mu,\sigma)}[f(\psi)]$ using LAX.

Our primary metrics for comparison are the objective function value obtained from the optimization, and the number of simulator function calls needed to find a minimum. The latter metric assumes that the simulator calls are driving the computation time of the optimization. We tune the hyper-parameters of all baselines for their best performance. For all experiments we run L-GSO and baselines ten times with different random seeds and show the averages and standard deviations in Figures 2 and 4. We use the same GAN architecture with $\sim 60k$ parameters for all experiments (for model details see Appendix A). When presented, true gradients of the objective function are calculated with PyTorch [46] and Pyro [9].

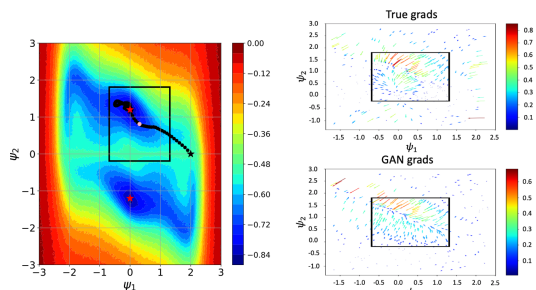

Figure 3: (Left) objective function surface of the "hump model" overlaid by the optimization path. Red stars are the objective optimal values. (Right) True gradients and GAN gradients, calculated at the yellow point. Black rectangles correspond to the current $\epsilon$ neighborhood around yellow point. Full path animation is available at `https://doi.org/10.6084/m9.figshare.9944597.v3`.

**Results** To illustrate L-GSO, we show the optimization path in $\psi$ space for three hump problem in Figure 3. We also show gradient vector fields of the true model and of the GAN surrogate estimation at random $\psi$ points during optimization, showing the successful approximation of the gradients by the surrogate. Visually, the true and the surrogate gradients closely match each other inside the surrogate training neighborhood (black rectangle).

The objective value as a function of the number of simulator calls in three experiments is seen in Figure 2. L-GSO outperforms score function based algorithms in speed of convergence by up to an order of magnitude in some problems. L-GSO also attains the same optima as other methods and the speed of convergence is comparable to numerical optimization. In Figure 2a, BO-RBF converges faster than all other methods, which is not surprising in such a low-dimensional problem. Conversely, in Figure 2b BO methods struggle to find the optimum due to the high curvature of the objective function, whereas the convergence speed of L-GSO is on par with numerical optimization. In general, L-GSO has several advantages over BO: (a) it is able to perform optimization without specification of the search space [27, 54], (b) the algorithm is embarrassingly parallelizable, though it should be noted that BO parallelization is an active area of research [63].

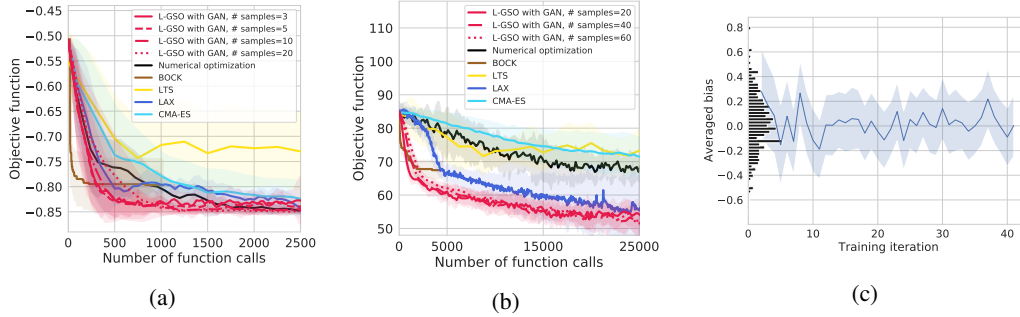

(a)           (b)           (c)

Figure 4: The objective function value as a function of the accumulated number of simulator calls for (a) Nonlinear Submanifold Three Hump problem, $\psi \in \mathbb{R}^{40}$, (b) Neural Network Weights Optimization problem , $\psi \in \mathbb{R}^{91}$. Shaded region corresponds to $1\sigma$ confidence intervals. (c) The bias (solid line) and one standard deviation (shaded region) of the GAN based L-GSO gradient averaged over all $\psi$ dimensions in the 10D Rosenbrock problem versus training step. Gray histogram shows the empirical bias distribution over all training iterations.

The bias and variance of the GAN based L-GSO gradient estimate averaged over all parameters for the 10D Rosenbrock problem for each training step can be seen in Figure 4c. The bias is calculated as the average difference between the true gradient and trained surrogates gradients, where each surrogate is trained with independent initialization and parameter sets $\{\psi'\}$ in the $\epsilon$-neighborhood of the current value $\psi$, for each training iteration. The variance is similarly computed over the set of trained surrogates gradients. The bias remains close to, and well within one standard deviation of, zero across the entire training. Bias and variance for each parameter, and additional details, can be found in Appendix B.

The benefits of L-GSO can further be seen in problems with parameter submanifolds, i.e., the Submanifold Rosenbrock, Nonlinear Submanifold Hump Problem and Neural Network Weights Optimization problems where the relevant $\psi$ parameters live on a latent low-dimensional manifold. No prior knowledge of the submanifold is used in the training and all dimensions of $\psi$ are treated equally for all algorithms. The objective value versus the number of simulator calls can be seen in Figures 2c and 4 where we observe that L-GSO outperforms all baseline methods. We also observe that BO methods were frequently not able to converge on such problems.

Figure 4a and 4b also show L-GSO with differing numbers of parameter space points used per surrogate training. In the submanifold problems, L-GSO converges fastest with far fewer parameter points than the full dimension of the parameter space. This indicates that the surrogate is learning about the latent manifold of the data, rather than needing to fully sample the volume around a current parameter point. The conditional generative network appears to learn the correlations between different dimensions of $\psi$ and is also able to interpolate between different sampled points. This allows the algorithm to obtain useful gradients in $\psi$, while using far fewer samples than numerical differentiation.

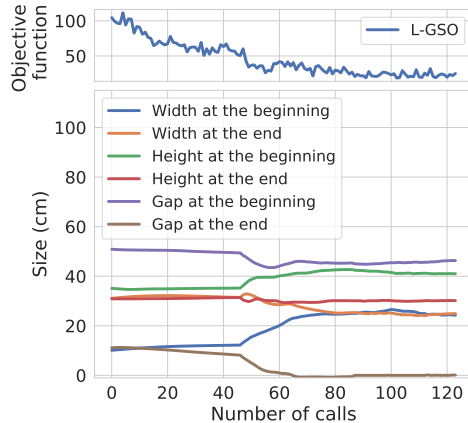

Figure 5: Magnet objective function (top) and six $\psi$ parameters (bottom) during optimization with L-GSO.

In terms of wall-clock run time, one L-GSO training and gradient descent step (Algorithm 1 steps 3 to 12) for the NN Weights Optimization problem took ~2 minutes. The full optimization totaled ~12 hours (of which nearly all time was used by the GAN training) and required sampling 23k points from the simulator. The BOCK run time was ~113 hours after using only 4k samples. LTS / LAX / CMA-ES were faster to converge in terms of run time (< 1 hour) on toy problems that were not simulation-time dominated. For problems

dominated by simulation time, the longer training time for L-GSO relative to LTS / LAX / CMA-ES was less consequential than the number of needed simulator calls (as in the physics example below).

## 4.1 Physics experiment example

We apply L-GSO to optimize the parameters of an apparatus in a real physics experiment setting that uses the physics simulation software GEANT4 [2] and FairRoot [3]. In this example, muon particles pass through a multi-stage steel magnet and their coordinates are recorded when muons leave the magnet volume if they pass through the sensitive area of a detection apparatus. As muons are unwanted in the experiment, the objective is to minimize number of recorded muons by varying the geometry of the magnet.

**Problem Definition** Inputs $\boldsymbol{x}$ correspond to the properties of incoming muons, namely the momentum (P), the azimuthal ($\phi$) and polar ($\theta$) angles with respect to the incoming $z$-axis, the charge $Q$, and $x$-$y$-$z$ coordinate $C$. The observed output $\boldsymbol{y}$ is the muon coordinates on the plane transverse to the $z$-axis as measured by the detector. The parameter $\boldsymbol{\psi} \in \mathbb{R}^{42}$ represents the magnet geometry. The objective function to penalize muons hitting the detector is

$$\mathcal{R}(\mathbf{y}; \boldsymbol{\alpha}) = \sum_{i=1}^{N} \left( \mathbb{1}_{Q_i=-1} \sqrt{(\alpha_1 - (\boldsymbol{y}_i + \alpha_2))/\alpha_1} + \mathbb{1}_{Q_i=1} \sqrt{(\alpha_1 + (\boldsymbol{y}_i - \alpha_2))/\alpha_1} \right) ,$$

where $\mathbb{1}$ is the indicator function, and $\alpha_1 = 5.6$ m and $\alpha_2 = 3$ m define the detector sensitive region.

**Results** of the optimization using L-GSO with a Cramer GAN [8] surrogate are presented in Figure 5. A previous optimization of this magnet system was performed using BO with Gaussian processes with RBF kernels [6]. The L-GSO optima has an objective function value approximately 25% smaller than the BO solution, while using approximately the same budget of ∼5,000 simulator calls. Total optimization time was ∼300 hours, of which ∼ 70% was simulation time. The L-GSO solution is shorter and has lower mass than the BO solution, which can both improve efficacy of the experiment and significantly reduce cost. More details can be found in the Appendix D.

## 5   Conclusions

We present a novel approach for the optimization of stochastic non-differentiable simulators. Our proposed algorithm is based on deep generative surrogates successively trained in local neighborhoods of parameter space during parameter optimization. We compare against baselines including methods based on score function gradient estimators [52, 24], numerical differentiation, and Bayesian optimization with Gaussian processes [43]. Our method, L-GSO, is generally comparable to baselines in terms of speed of convergence, but is observed to excel in performance where simulator parameters lie on a latent low-dimensional submanifold of the whole parameter space. L-GSO is parallelizable, and has a range of advantages including low variance of gradient estimates, scalability to high dimensions, and applicability for optimization on high curvature objective function surfaces. We performed experiments on a range of toy problems and a real high-energy physics simulator. Our results improved on previous optimizations obtained with Bayesian optimization, thus showing the successful optimization of a complex stochastic system with a user-defined objective function.

## Broader Impact

This work presents a method for black-box optimization, targeting situations where the black box is a stochastic simulator that is costly to evaluate. This work could impact scientific and engineering disciplines, which frequently need to optimize objectives represented by simulators, for instance for experiment design and design optimization. Such types of optimization often arise in fields including physics, biology, and chemistry. Speeding up such computations could lead to faster iteration of optimization cycles and reduce human intervention, thus providing faster discoveries and more efficient design of experimental apparatus. Any biases present in the simulator will also likely be present in the learned surrogate, and therefore could lead to negative outcomes. Careful analysis of the simulator and the optimization solution is required by domain experts to avoid any negative outcomes.

## Acknowledgments and Disclosure of Funding

We would like to thank Gilles Louppe and Auralee Edelen for their feedback, and Fedor Ratnikov for the fruitful discussions. SS is supported by the Imperial College President's PhD scholarship. MK is supported by the US Department of Energy (DOE) under grant DE-AC02-76SF00515 and by the SLAC Panofsky Fellowship. AU and VB are supported by the Russian Science Foundation under grant agreement n° 19-71-30020 for their work on Bayesian Optimization and FFJORD methods. AGB is supported by EPSRC/MURI grant EP/N019474/1 and by Lawrence Berkeley National Lab. The reported study utilized the supercomputer resources of the National Research University Higher School of Economics.

## Footnotes

[2] A detailed study of alternative sampling techniques is left for future work.

[3] For a deterministic simulator this parameter could be chosen proportionally to learning rate. If the objective function is stochastic, one might want to choose $\epsilon$ big enough so that $\mathbb{E}\left|\mathcal{R}(\mathbf{y}_{\psi-\epsilon}) - \mathcal{R}(\mathbf{y}_{\psi+\epsilon})\right)| > \text{Var}(\mathcal{R}(\mathbf{y}_\psi))$.

[4]The orthogonal mixing matrix $A$ is generated via a QR-decomposition of a random matrix sampled from the normal distribution.

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
