[Supplementary Material]

# Black-Box Optimization with Local Generative Surrogates Supplementary Material

## A    Surrogates Implementation Details

### A.1    GAN Implementation

For the training of the GANs we have used conditional generative network, with three hidden layers of size 100 and conditional discriminative network with two hidden layers of size 100. For all the hidden layers except the last one we have used $tanh$ activation. For the last hidden layer $leaky\_relu$ was used. The conditioning is performed via concatenating the input noise $z$ with input parameters $\psi$. The learning rate and batch size is set to $0.0008$ and $512$ correspondingly. We have used the idea from the [57] to adjust the learning rate and a batch size for optimal training speed and performance. We have used Adam optimizer for both discriminator and generator with $\beta_1 = 0.5, \beta_2 = 0.999$. We have trained GAN only for 15 epochs for all the experiments. The number of discriminator updates per one generator update is $n_d = 5$. In case of the Cramer GAN [8] we have used gradient penalty with $\lambda = 10$, discriminator output size equal to 256 and the number of discriminator updates $n_d$ is set to 1.

### A.2    FFJORD Implementation

Training procedure and architecture of FFJORD model [23] were fixed for all experiments. We have used two hidden layers with 32 neurons each. The learning rate and batch size are set to $10^{-3}$ and 262144 respectively. It was trained with SWATS optimizer [33] until convergence, i.e. until log-likelihood is no longer improves for more than 200 epochs. For all hidden layers *tanh* was used as nonlinearity, batch normalization lag were set to $10^3$ and *fixed_adam* were used as ODE solver. Usage of adaptive ODE solver and/or more elaborate choice of architecture probably could improve performance of the algorithm, but, firstly, it is out of the scope of our work, and, secondly, we were aiming to show that even without tuning for specific problem algorithm could shows performance comparable with recent works.

Original version of FFJORD does not have a support of conditional input. To address this issue we rewrote one of the base layers that were used in FFJORD library. We have added additional two-layers network with hidden dimensionality equal 8 that takes as an input conditional information and injects it in base layer output as an additive bias term.

### A.3    Monitoring quality of the surrogate

During optimization we are constantly monitoring various statistics between samples from simulator and surrogate. The example of such statistics is presented in Fig 6. This is done to ensure that the surrogate learn a meaningful approximation of the simulator on each iteration of optimization and if this is not the case, the user can further tune the model.

Figure 6: An example of monitored statistics during surrogate training for one iteration of optimization. Left to right: Jensen–Shannon divergence, Kolmogorov–Smirnov statistic, difference between order one, two and three moment of the distributions from simulator and surrogate.

# B  Bias of the estimator

The gradient bias is calculated per component of the gradient vector, i.e., if $\boldsymbol{\psi} \in \mathbb{R}^d$, then we present bias per component of this vector at each point of the optimization step. We calculate bias vector as follows:

---

**Algorithm 2** Procedure to estimate the bias of the L-GSO

---

**Require:** number N of $\boldsymbol{\psi}$, number M of $\boldsymbol{x}$ for surrogate training, number R of gradient estimates at point $\boldsymbol{\psi}_t$, trust region $U_\epsilon$, size of the neighborhood $\epsilon$, Euclidean distance $d$
1: **for** t = 1, ..., T **do**
2:    $\mathbf{b} \leftarrow \varnothing$
3:    **for** r = 1, ..., R **do**
4:       Sample $\boldsymbol{\psi}_i$ in the region $U_\epsilon^{\boldsymbol{\psi}_t}$,
          $i = 1, \ldots, N$
6:       Sample training data $(\boldsymbol{y}_j^i, \boldsymbol{x}_j^i, \boldsymbol{\psi}_i)$,
          $j = 0, \ldots, M$
8:       Train generative model $S_\theta(\boldsymbol{z}_l, \boldsymbol{x}_l, \boldsymbol{\psi}_l)$,
          $\boldsymbol{z}_l \sim \mathcal{N}(0,1), l = 1, \ldots, MN$
10:       Compute $\nabla_{\boldsymbol{\psi}|\boldsymbol{\psi}_t} \mathbb{E}[\mathcal{R}(\bar{\boldsymbol{y}})]$ from surrogate
11:       $\mathbf{b_r} \leftarrow \nabla_{\boldsymbol{\psi}|\boldsymbol{\psi}_t} \mathcal{R}(\boldsymbol{y}) - \nabla_{\boldsymbol{\psi}|\boldsymbol{\psi}_t} \mathbb{E}[\mathcal{R}(\bar{\boldsymbol{y}})]$
12:    **end for**
13:    $\mathbf{bias}_t = \frac{1}{R} \sum_{r=1}^{R} \mathbf{b}_r$
14:    $\mathbf{variance}_t = \frac{1}{R-1} \sum_{r=1}^{R} (\mathbf{b}_r - \mathbf{bias}_t)^2$
15: **end for**

---

Figure 7: The bias (solid line) and one standard deviation (shaded region) of the GAN based L-GSO gradient estimate for all dimensions of $\boldsymbol{\psi}$ of the 10D Rosenbrock problem is shown as a function of the training step. The gray histograms shows the empirical distribution of bias averaged over all training iterations.

The bias and variance for all parameters in the 10 dimensional Rosenbrock problem are presented in the Figure 7.

# C  Optimization Implementation Details

Throughout all the experiments Adam [34] optimizer with default hyperparameters and learning rate equal $10^{-1}$ was used to perform update of the $\boldsymbol{\psi}$ parameters.

Latin Hypercube sampling window of size $\epsilon = 0.2$ was used for the "Rosenbrock", "Submanifold Rosenbrock", "Nonlinear Submanifold Hump", and "Neural Network Weights Optimization" problems, $\epsilon = 0.5$ was used for "Three Hump" problem.

## C.1  Procedure For Mixing Matrix Generation

10-dimensional mixing matrix $A$ could be generated with the following Python code:

```
import numpy as np
def generate_orthogonal(in_dim, out_dim, seed=1337):
    assert in_dim > out_dim
    mixing_covar, _ = np.linalg.qr(np.random.randn(n_dim,out_dim))
    return mixing_covar
```

## C.2  Procedure For Initialization of Neural Network For Boston Regression Problem

Neural network for Boston regression problem initialized as a two-layer network with tanh-nonlinearity with predefined weights, using PyTorch.

```
import torch
from torch import nn
def make_boston_net():
    torch.manual_seed(1337)
    net = nn.Sequential(
    nn.Linear(13, 6),
    nn.Tanh(),
    nn.Linear(6, 1)
    )
    initial_weights = torch.tensor(
        [0.0215, 0.0763, 0.0879, 0.0102,
        0.095, 0.0508, 0.088, 0.101,
        0.0782, 0.0684, 0.0658, 0.0509,
        0.0207, 0.0618, 0.0756, 0.00784,
        0.0968, 0.0685, 0.0113, 0.0745,
        0.00154, 0.0772, 0.0472, 0.000906,
        0.0723, 0.0779, 0.0594, 0.0785,
        0.0918, 0.0634, 0.0853, 0.105,
        0.00407, 0.0789, 0.0035, 0.0581,
        0.0375, 0.0632, 0.0669, 0.00293,
        0.0901, 0.0208, 0.0388, 0.0893,
        0.00104, 0.0598, 0.0745, 0.08,
        0.0283, 0.0106, 0.0371, 0.0667,
        0.0331, 0.0356, 0.0661, 0.0554,
        0.084, 0.0398, 0.00286, 0.0281,
        0.0246, 0.0208, 0.0358, 0.033,
        0.0421, 0.0505, 0.00544, 0.0269,
        0.00527, 0.0569, 0.00538, 0.0786,
        0.102, 0.0452, 0.0444, 0.105,
        0.0765, 0.0689, 0.0249, 0.0933,
        0.037, 0.0762, 0.0882, 0.0505,
        0.0688, 0.0666, 0.101, 0.0857,
        0.0488, 0.0303, 22.5328])
    net[0].weight.data = initial_weights[: 6 * 13].view(6, 13).detach
        ().clone().float()
    net[0].bias.data = initial_weights[6 * 13: 6 * 13 + 6].view(6).
        detach().clone().float()

    net[2].weight.data = initial_weights[6 * 13 + 6: 6 * 13 + 6 + 6].
        view(1, 6).detach().clone().float()
    net[2].bias.data = initial_weights[6 * 13 + 6 + 6: 6 * 13 + 6 + 6
        + 1].view(1).detach().clone().float()
    net.requires_grad_(False)
    return net
```

### C.3   Numerical Derivatives

To obtain numerical derivatives of $\mathcal{R}$ we are using central difference scheme:

$$f'_{\psi_i} \approx \left(\bar{\mathcal{R}}(\psi_1, \ldots, \psi_i + \epsilon, \ldots, \psi_p) - \bar{\mathcal{R}}(\psi_1, \ldots, \psi_i - \epsilon, \ldots, \psi_p)\right)/2\epsilon, \tag{7}$$

Where, $\bar{\mathcal{R}} = \frac{1}{N}\sum_{i=1}^{N}\mathcal{R}(F(z_i, x_i; \boldsymbol{\psi}))$, $x_i \sim p(\mathbf{X})$, $z_i \sim p(\mathbf{Z})$. For all experiments we set $\epsilon = 0.1$.

We can not use small $\epsilon$ due to the stochastic nature of $\bar{\mathcal{R}}$ (see appendix E, where we compare results with different values of $\epsilon$).

## D   Details of the Physics Problem

Muons are bent by the magnetic field and simultaneously experience stochastic scattering as they pass through the magnet which causes random variations in their trajectories. The coordinates

Table 1: Comparison of the optima, obtained by L-GSO and Bayesian optimization for the physics problem.

| Algorithm | Objective value | Magnet length (m) | Magnet weight (kt) |
|-----------|-----------------|-------------------|--------------------|
| L-GSO | $\sim 2200$ | 33.39 | 1.05 |
| BO | $\sim 3000$ | 35.44 | 1.27 |

Figure 8: The $x$-$z$ axes and $y$-$z$ axes profiles of the magnet system (the post optimization shape is shown). Animation of optimization process is available at `https://doi.org/10.6084/m9.figshare.11778684.v1`.

Figure 9: Muon hits distribution in the detection apparatus (depicted as red contour) obtained by Bayesian optimization (Left) and by L-GSO (Right), showing better distribution. Color represents number of the hits in a bin.

perpendicular to incoming direction (the z-axis in Figure 8) are recorded. The loss function is constructed in such a way that muons with positive charge should be swept by the magnet along the negative $x$-direction while those with negative charge should be swept to the positive $x$-direction. Overall the loss encourages muons to be bent outside of the detector area (red lines in Figure 9). The magnet is constructed from six trapezoidal shapes with gaps each of which is described by seven parameters, as presented in Figure 8. Thus, for this task $\psi \in \mathbb{R}^{42}$. Mathematically, formulation is $X = \{P, \phi, \theta, Q, C\}, X \in \mathbb{R}^7, \ y \sim \mathbb{R}^2$ where $y$ is a simulator output representing hit coordinates in the sensitive detector. X is sampled from empirical distribution $H$ (histogram), produced upfront. To make our optimization comparable with previously applied BO optimization, during optimization we have been working with subsample of $H$ of size of order of $O(500,000)$ events, same as in case of BO application. The objective function value reported in the Figure 5 is calculated on this sample. To perform cross-validation of the obtained optima, we run physics simulation on the largest available sample, which does not contain samples from $H$. We have also validated the BO optima on the same available sample. The comparison is presented in Table 1. Both BO and L-GSO have been compared on the simplified geometry of the experiment. The distributions of muons in the detection apparatus obtained by L-GSO is compared with BO optimization in Figure 9.

# E   Grid Search of Optimal Parameters

## E.1   Grid Search Hyperparameters For L-GSO

We have optimized crucial hyperparameters of L-GSO, such as learning rate, size of the sampling neighborhood $\epsilon$ and the number of samples of $\psi$ in this neighborhood with grid search. The grid search for Three hump and Rosenbrock problem is presented in Figure 10a, 10b. As it can be seen, for both problems best quality is obtained when number of samples is approximately equal to the dimensionality of the problem and when learning rate is close to 0.1. We found that learning rate 0.1 is optimal for all the toy problems under consideration. Thus, we have fixed it to be 0.1 for other grid search experiments. In the Figure 11a, 11b we present the grid search for 100 dimensional Degenerate Rosenbrock problem for number of samples per iteration and size of the neighborhood. We found that L-GSO is very sensitive to the size of the neighborhood $\epsilon$, whereas surprisingly robust to the number of samples, as it is seen in Figure 11a.

(a)

(b)

Figure 10: Grid search of learning rate and number of samples for L-GSO. Color represents final quality for Three hump problem (Left) and for Rosenbrock problem (Right).

(a) Final value of objective function $\mathcal{R}$ of L-GSO for 100 dimensional Degenerate Rosenbrock problem.

(b) Number of samples (calls of the simulator) needed by L-GSO to converge to final value of objective funciton.

Figure 11

## E.2   Grid Search Hyperparameters For Numerical Differentiation

We performed grid search over the order of numerical scheme $n$ and step size $h$ for numerical optimization for all four toy problems. As an example, the results for the toys problems are presented in Figure 12a, 12b, 13a, 13b, 14a, 14b.

(a) Final value if objective function $\mathcal{R}$ of numerical differentiation for hump problem.

(b) Number of samples(calls to the simulator) needed by numerical differences to converge.

Figure 12

(a) Final value if objective function $\mathcal{R}$ of numerical differentiation for 10-dim Rosenbrock problem.

(b) Number of samples(calls of the simulator) needed by numerical differentiation to converge.

Figure 13

(a) Final value if objective function $\mathcal{R}$ of numerical differentiation for 100-dim Degenerate Rosenbrock problem.

(b) Number of samples(calls of the simulator) needed by numerical differentiation to converge.

Figure 14