[Reviews · NeurIPS 2020]

Review 1

Summary and Contributions: The authors propose a novel method (L-GSO) for black-box optimisation using local neural network surrogates, aiming to minimise the number of simulator calls. L-GSO compares favourably empirically to alternative techniques on a number of problems in terms of convergence.

Strengths: This paper introduces a potentially powerful new way to solve black-box optimisation problems. The proposed method is evaluated on a number of well-motivated problems. Hyperparameter choices are discussed and made transparent. With some additional work (see weaknesses) this paper could serve as a basis for thorough empirical comparison of various black-box optimisation techniques.

Weaknesses: While I was impressed by the breadth of problems, my main criticism concerns the evaluation of baselines: While several relevant baselines are included for the Probabilistic Three Hump Problem and (Submanifold) Rosenbrock Problems, they are missing for the Submanifold Hump problem, the Neural Network Weights Optimization problem, and the Physics experiment example (e.g. LTS and Guided evolutionary strategies). To better judge the strength of empirical results, these baselines should be included on all problems. Assuming that the simulation cost is dominant for optimisation problems of interest is a reasonable assumption. Nevertheless, total runtime should be reported -- runtime information is important for reproducibility and might be of practical interest as well. In the current form of the algorithm, simulator parameters psi are choosen randomly using Latin Hypercubes. In order to truely be efficient in the number of simulator calls, adaptive/active learning schemes for choosing parameters psi would likely be able to greatly improve efficiency of the proposed algorithm. It would have strengthened the paper to see initial experiments in this direction as part of the manuscript. Update following author response: My main criticism with the paper concerned the evaluation of baselines. This is addressed in the authors response. While I don’t find the argument that baselines were originally excluded merely for reasons of visual clarity entirely convincing (the authors could just have added a supplementary figure; baselines are crucial for empirical evaluation), it’s good that they will be included in the next version of the manuscript. In addition, I appreciated that details regarding runtimes were provided, as requested. Given that 2 out of 3 raised weaknesses were addressed, I am raising my score from below to above acceptance threshold.

Correctness: Apart from including baselines for all problems, error bands should be included for all methods: At the moment, errors are only reported for L-GSO. In Figure 3b, the error band for L-GSO with GAN appears to be missing.

Clarity: Overall, the paper is well written and easy to follow. Two points that would deserve further clarification are: 1) How do the authors deal with training the surrogate model: is the surrogate model initialized from scratch for every gradient step or is some sort of continual learning used? 2) It is stated that the method assumes "that the objective function is continuous and differentiable [...]" but that the "assumption may be relaxed by incorporating the objective into the surrogate". Could the authors clarify this further, i.e., how would this look in practice?

Relation to Prior Work: Prior work seems appropriately discussed, judged by my somewhat limited knowledge of the optimisation literature. In many instances, the authors strongly constrast their approach to Bayesian Optimisation, which typically does not use gradient information. However, BO variants that can utilise gradient information have been proposed (e.g. Wu et al., 2018, Bayesian Optimisation with Gradients). Wouldn't it be feasible to likewise use the authors method of obtaining surrogate gradients in a BO framework?

Reproducibility: Yes

Additional Feedback:


Review 2

Summary and Contributions: This work proposes a novel method for doing black box optimization by building differentiable local surrogate models and doing gradient based optimization on these surrogates. They show that doing this allows for accurate optimization while showing benefits in regions where the parameter space is constrained to a low dimensional manifold.

Strengths: Paper considers a very relevant problem, widely applicable in the sciences and even in domains like robotics. The algorithm proposed is simple, seems to work well on certain problem types (with lower intrinsic dimensionality), and has been tested pretty thoroughly. I think the description is generally clear, the experimental analysis mixes simple and more complex domains and the results are compelling.

Weaknesses: There seems to be a lack of a very clear reason (intuitively or theoretically) why this algorithm will work well when the same samples are used for training a generative model (with no oracle gradient information), and a different black box optimization algorithm. Also a bit of clarification in the methods section should be addressed to make this paper stronger.

Correctness: Methodology seems correct.

Clarity: Overall the paper is pretty clear and well written.

Relation to Prior Work: Yes this has been discussed pretty clearly, although some things should be moved to the related work section rather than be left in other sections like experiments.

Reproducibility: Yes

Additional Feedback: Introduction: The focus of this work seems to be on cases where the inputs are stochastic and the simulator are stochastic. This would be equally applicable to scenarios which were deterministic but otherwise non differentiable right? One related work that comes to mind is related to sobolev training. It’s a bit different in motivation and setup but might be nice to cite. The introduction is generally well motivated and concise. One question that pops up from the introduction is - if we use Monte Carlo samples to actually train the generative model, and then use the same generative model for gradients, where is the extra juice coming from? If there are no extra assumptions or extra samples, wouldn’t the gradients of the generative model contain essentially exactly the same amount of information as directly using the Monte Carlo samples for estimation. I would guess that something about the generative model makes these gradients more informative or something despite being trained on the same set of Monte Carlo samples? Method: Same question as above. If the samples are really the only source of *ground truth* information, then why would building a generative model for the same actually help? It doesn’t contain any more information about the true model and the gradients are equally arbitrary right? Is it a better way to inject priors, is optimization better conditioned? This should be very clearly addressed and explained early on. It makes sense to estimate the model locally around the point, but there are no guarantees the gradients are particularly sensible for a small number off samples right? Are there any theoretical guarantees that can be provided for the algorithm with number of samples or for the trust region described in lines 118 - 128? Also how slow is the retraining of local surrogates as opposed to upfront training or black box optimization in terms of wall clock time? Lines 146 - 149, these seem crucial. It explains in what case the proposed method works better than numerical methods. Why does the low intrinsic dimensionality of the parameters actually allow the method to do that much better? Can this be elaborated on some more. Also can some examples of this scenario be provided? One thought might be - if the algorithm explicitly trains to match gradients of the generative model with either a few available gradient signals or more expensively computed numerical gradients, might that also help ensuring the gradients of the generative model are accurate? Another question might be - how does this change with regularization and complexity of the generative model? If \theta is very very high dimensional how does this affect the gradient signal and optimization? Also how would we deal with really high dimensional \psi simulator parameters? The generative model would become really really complicated then perhaps? Overall the method makes a lot of intuitive sense, and I think the description is fairly clear. Some of the details on page 3 (end) and page 4 should go into the experiments section as details rather than the methods section. I think my main question is the one about where the extra information/signal is coming from if the generative model/gradients are all trained from the same signal as we would use for direct optimization? Any theoretical guarantees that can be provided? Related work: Backprop through the void [grathwohl] seems quite relevant, should this be discussed in this section? It seems like this is discussed later but it should be brought up here. Results: Fig 3 results are quite convincing and show that while other methods can in some cases be comparable but in the low intrinsic dimensionality, but high extrinsic dimensionality cases the algorithm does perform a fair bit better. The figure 2 gradients between GAN and true look a fair bit different to me though, which makes it a bit surprising that the algorithm works. What insights do the authors have on this? The authors also plot bias and variance in Fig 4, which sheds some light on the gradients generally having low bias which might explain good performance. Does the variance change across different problems? Or is it fairly consistent across all considered domains/model classes? The physics experiment is very cool! I wonder how much lower the values can get if the LGSO algorithm is allowed to run for longer. How much does the performance of the algorithm vary as the simulator behavior is changed (stochasticity, dimensionality, etc), and across problem domains?? It might also be cool to show applicability on a different domain like robotics/planning or something that uses a simulator. Overall, I find the idea quite intuitive and appealing. I still don’t quite understand where the juice comes from in squeezing out more information from the same number of samples, and getting the gradients to be informative. But if the authors can address that then I think the paper is quite good, and there is a good mix of toy experiments, analysis and a real world physics experiment. It would have been nice to get a bit of theory/guarantees about the algorithm as well, but I don’t think it is by itself a disqualifying factor from publication.


Review 3

Summary and Contributions: The authors present a novel method, called L-GSO, for the global optimisation of stochastic, non-differentiable ‘black box’ simulators. Their method trains deep generative surrogate models, such as a GAN or a flow, within local regions of the parameter space. Using gradients obtained through these surrogate models, e.g. via back-propagation, and a problem/domain-specific objective function, they can perform SGD on the model parameters. The authors compare their method against several baselines, such as Bayesian optimisation, and find that their method performs similarly, or better. When the model parameters lie on a lower-dimensional manifold, their method exceeds other work in terms of function evaluations and, sometimes, objective function value.

Strengths: The main idea of only considering local parameter space, training a generative model and then using its gradient for SGD on the simulator parameters is simple, yet appears surprisingly powerful. There are many benefits of their method, such as being able to re-use previously sampled data points and deal with objective surfaces that have a high curvature. Furthermore, their method allows for the inclusion of any generative model; the authors have used GANs and FFJORD models, both of which are considered state-of-the-art. I believe the authors might have under-selled the potential scalability of their method. Other ‘black box’ optimisation algorithms, such as Bayesian optimisation (even with a cylindrical kernel), have severe scalability issues in regards to the number of parameters, which their method does not have, due to using SGD on the model parameters. I’m uncertain, however, about how well the training of generative models scales with the dimensions of psi (a short comment on that in the main text would be helpful). The authors set their method apart nicely from previous work, through a thorough literature review and by comparing their method to plenty of baselines in their experiments.

Weaknesses: Because they build local surrogate models, not global ones, and they used SGD on the model parameters, L-GSO might very well be stuck in a local optima. Other methods with a more pronounced exploration-exploitation trade-off, e.g. Bayesian Optimisation, do not necessarily have this issue. This is naturally a common problem in machine-learning but because other methods might not experience this, it would probably be worth noting. It appears to me that in line 11 of Algorithm 1, their method requires them to know the gradient of the objective function R with respect to the data y. Is that correct? If so, this might be true for a large class of use cases, but I suspect that sometimes this might not be the case. For instance, in domains such as Bayesian experimental design or representation learning the objective function might be e.g. the mutual information between simulated data and model parameters. In that case, that required gradient would not be straightforward to compute. Perhaps the authors could clarify this point and make a note that this gradient needs to be known. The authors note that a natural drawback of using deep generative models, as supposed to simpler, linear models, is the selection of the model architecture. However, this is a common problem in machine-learning and so I don’t think this is of too much significance (especially because their method works with any generative model). ***EDIT*** The authors argued that they could incorporate the objective R into the surrogate model and relax the assumption that R must be differentiable w.r.t. y (they state this in the main text as well). This sounds feasible to me but I would have appreciated slightly more detail e.g. in the appendix.

Correctness: All claims made by the authors seem correct and the empirical methodology is sound as well.

Clarity: From a notational point-of-view, I think it would be helpful if the authors would write out the distributions over which the expectations are taken, in particular in Equations 1 and 2. While this may still be alright for Eqn 1, where they write down the integral that defines the expectation shortly afterwards, I believe it’s crucial to know this in Eqn 2. Within the main text, the authors refer to the current parameter (at which they are performing SGD) as \psi and parameters that are sampled inside the trust region as \psi’ (i.e. with a prime). In Algorithm 1 they drop the prime to indicate parameters samples inside the trust region, which might confuse readers. I would suggest to add the prime on the \psi_i in lines 3-6 of Algorithm 1. Regarding Section 4.1, it would improve clarity of exposition if the authors provided some intuition, or domain expert knowledge, about where the equation for the objective function comes from (either in the main text or, if longer, in the supplementary material). If I understood this correctly the authors say that they use a spherical trust region (see line 126 on p.3) but the trust region in the animation (see Figure 2 on p.5), which was very well done, appears rectangular. I’m not sure if I misunderstood the form of their trust region, or if there they used a different trust region in the animation. This leads me to another immediate question that I believe should have been addressed: Are there perhaps other trust regions (or difference measures between psi and psi’) that might help the optimisation, depending on the problem? If so, I believe this would be worth noting. ***EDIT*** The authors clarified that they did not use a spherical trust region, but a square one. They said that they would correct this.

Relation to Prior Work: The authors describe in detail how their work differs from other methods in Section 3.

Reproducibility: Yes

Additional Feedback: Regarding Figure 4: Would it be possible to run the training of L-GSO with a GAN surrogate on the 10D Rosenbrock problem for longer, or with more than 5 re-runs? While I believe the authors that the bias of the gradients is zero and the variance is (relatively) low, I feel like this plot can still be improved. Regarding Figure 6: I think the legend in the top plot can be dropped and the figure caption should say that these plots were made with L-GSO. Additionally, to me it seems like the six parameters in the bottom plot have not quite converged yet, same for the objective function. Would it be possible to run this experiment for a little longer to make sure that the parameters have converged? ***EDIT*** In their response the authors provided simulation times for the physics model and argued that rerunning for longer (or other baselines) would not be feasible in a short time, which seems reasonable.


Review 4

Summary and Contributions: In this work, the authors propose a gradient-based black-box optimization method based on generative models. The idea is to provide a gradient for black-box function by differentiating through a local generative surrogate. However, this idea is not novel in the literature. In a previous work [1], the authors proposed a GAN-based model with a continuous score for a desirable black-box metric(function), in which training the discriminator is learning of local surrogate. By setting the continuous score s=R(y), the method in [1] is quite similar to the proposed method.

Strengths: 1. The proposed method may cover a broader setting with a stochastic nature of the targeted black-box function, i.e. f= E(R(y)). It may be suitable for stochastic simulator optimization. 2. The authors formulated the problem in a more clear way compared with [1].

Weaknesses: 1. A closely related work is missing. By setting the continuous score s=R(y), then using the gradient of the surrogate (discriminator) to approximate the gradient, the method in [1] is quite similar to the proposed method. I would like to see a clear discussion about the relation to [1]. 2. The method in [1] can serve as a closely related baseline. I would like to see a comparison with this baseline. 3. All the comparisons with other baselines are performed on toy examples. I would like to see a comparison in a practical setting. ================================================================================= Thanks to the authors' feedback. Both the proposed method and [1] are GAN-based methods with local surrogates (see page 3 in [1]) to address black-box functions. From this perspective, I think they are closely related. Compared with [1], the proposed method can handle stochastic black-box functions instead of a fixed one, which is a positive side. The method in [1] is used for a fixed black-box function. Considering this, I decided to increase my score.

Correctness: The emperical studies are weak to support the claim. Futher comparisons with closely related baseline are missing.

Clarity: The paper is well written.

Relation to Prior Work: A closely related work is missing. In a previous work [1], the authors proposed a GAN-based model with a continuous score for a desirable black-box metric(function), in which training the discriminator is learning of local surrogate. By setting the continuous score s=R(y), the method in [1] is quite similar to the proposed method. [1] Fu et al. MetricGAN: Generative Adversarial Networks based Black-box Metric Scores Optimization for Speech Enhancement.

Reproducibility: Yes

Additional Feedback:

[Author Response · NeurIPS 2020]

We thank the reviewers for their thoughtful comments! We are encouraged that they find the empirical comparison of
our method, L-GSO, detailed and thorough, based on a mix of simple and complex well-motivated domain problems
(R1, R2, R3), noted broad applicability in the scientific domain and appreciated the physics experiment (R2), clear
hyperparameter choice (R1) and simplicity of the implementation, yet performing well on a range of problems (R2, R3).
We agree that we might have been under-selling scalability of our method (R3) w.r.t. parameters of the surrogate model,
though this was not our goal. We are glad the text is clearly written (R2, R4) with a broad literature review (R1, R2).

**(R1, R2) Scalability w.r.t. simulator parameters $\psi$ dimensionality:** From empirical observations, L-GSO generally
scales linearly with the dimensionality of $\psi$. However, in cases where $\psi \in \mathbb{R}^D$ lies on a manifold of dimension d lower
than the extrinsic dimension of the space D, L-GSO requires less than D samples for producing a reasonable gradient.
We note this on lines [146-149].

**(R2) Scalability w.r.t. surrogate parameters $\theta$ dimensionality:** As R3 noted, L-GSO scales easily with the dimen-
sionality of $\theta$ due to using SGD on model parameters. Also, our method is quite robust to the choice of the architecture,
as for all experiments simple and more complex, we have used an identical GAN architecture with 60k parameters.

**(R1) Adding baselines for Submanifold Hump and NN Weight Optimization problems:** All baselines have been
applied to these problems, but for visual clarity, we did not show comparisons in Fig. 5, which focuses on comparing
the speed of convergence w.r.t. to the num. of $\psi$ samples. Other methods did not perform as well, but can be added to
the figure. Similarly, for visual clarity, error bands were not included for all baselines in Fig. 2. We will add plots for all
baselines with respective error bands with an increased number of re-runs into appendix as requested.

**(R1, R3) Adding baselines and longer L-GSO training on physical problem:** For the physics experiment, simulation
calls are extremely costly; 120 steps took ~300h of which ~ 70% was simulation time. This physical example was a
proof-of-concept of applicability of our method for real physics problems. A comparison to BO is listed on line [339],
but rerunning other baselines is not feasible on short timescales.

**(R1, R2) Total runtime of the algorithm:** For NN Weights Optimization problem one epoch takes ~2m, totalling
~12h for training (on 23k samples) of which nearly all time was GAN training. BO runtime was 113h, using only
4k samples before termination. Other baselines (LTS/Void/GES) were faster(<1h), b/c the toy problems were not
simulator time dominated. For problems dominated by simulation time, the longer training time for L-GSO relative to
LTS/Void/GES is less consequential than the number of needed simulator calls (as in the physics example).

**(R1, R3) Adaptive/active learning for sampling process/other trust regions types:** We agree, adaptive/active
learning is a key next step. We thought it important to first establish the method, even w/o adaptive sampling schemes.
We also thank R3 for pointing out the sampling region formula typo (trust region is square).

**(R1, R2) Objective function R must be differentiable w.r.t. y:** It is correct, however on line [61-63] we note that this
assumption can be relaxed by incorporating the objective into the simulator, i.e. re-assigning $y' = R(y)$ and $R' = y'$.

**(R1) Is the surrogate model initialized from scratch for every gradient step?** Yes, this is discussed on line [127].

**(R1, R3) Combining Bayesian Optimization with our approach:** We agree, gradient information from our surrogate
could be used in down-stream optimization like BO, but we have not investigated this yet. This could also improve the
exploration vs. exploitation, as SGD focuses on exploitation.

**(R2) Are there any theoretical convergence guarantees?:** We do not have explicit theoretical guarantees. However,
we empirically observe unbiased gradients in Fig. 4. We can appeal to the theory of SGD convergence which state that,
subject to some conditions, it is guaranteed to converge (arXiv:1805.08114). We will discuss this in the revised paper.

**(R2) How does the generative model help compute gradients?:** The extra assumptions come from defining a
model class (the gen model) that restricts the set of potential solutions and allows interpolation. The gen model is
building a continuous approximation of the simulator distribution from the samples and thus implicitly regularizing
the solution, which is not done with numerical methods. So the gen model has more information in terms of the
implicit prior defined by choice of model and optimization scheme. We believe this relates to recent work such as
[arXiv:1809.04542,arXiv:1811.03259] on inductive bias and generalization in deep generative models in GANs. On
lines [207-215], we also briefly discuss that our findings on the effectiveness of our method in case of low intrinsic
dimensionality are consistent with papers on adaptivity of deep learning models to intrinsic dimensionality. On matching
/ using gradients, this could indeed be helpful to incorporate them when available.

**(R4) Novelty w.r.t. MetricGAN:** We respectfully disagree with the reviewer, while MetricGAN (MG) is related (and
can be noted in related work), it is not directly applicable to black box optimization (BBO). MG aims to find optimal
parameters of the generator network while approximating a fixed black box metric (function) with a regression-based
surrogate, while we aim to optimize the black box function itself, which are quite different tasks needing different
algorithm considerations. In no place does MG optimize over simulator parameters to generate better observations, it
fits to fixed observations. MG could potentially be adjusted for the BBO setting, but this is clearly outside of the scope
of the existing MG work, and we can not be expected to undertake R&D to make MG applicable to this setting. We also
disagree with the notion that empirical studies, which compare to a variety of baseline models and show favourable
performance, is weak based solely on not comparing to MG which is not directly applicable without further research.
Please consider reviewing the work more holistically and in the broader setting of established BBO methods, rather
than only through the lens of a single additional method which is not directly applicable.

[Meta-Review · NeurIPS 2020]

Good paper that employs generative models for optimizing black-box objectives, where Bayesian optimization might otherwise be used. The insight that the generative model need only be locally accurate to provide a descent direction for optimization is interesting. Overall, the reviewers agree that the paper has merit, and that the proposed method is simple and powerful. The reviewers made several suggestions for improvement. I would encourage the authors to take to heart the reviewers' suggestions when preparing the camera-ready, and to add the additional details (e.g. baselines, wall-clock times) that the reviewers requested.